The impact of pneumococcal vaccination on pneumonia mortality among the elderly in Japan: a difference-in-difference study

Jung Sung-mok
Lee Hyojung
http://orcid.org/0000-0003-0941-8537 Nishiura Hiroshi nishiurah@gmail.com
Graduate School of Medicine, Hokkaido University , Sapporo, Hokkaido , Japan
Tulkens Paul
Electronic publication date: 2018 Dec 12
Publication date: 2018
Volume: 6
Electronic Location ID: e6085
Received 2018 Aug 30; Accepted 2018 Nov 6
Copyright: © 2018 Jung et al.
Copyright year: 2018
Copyright holder: Jung et al.
License: This is an open access article distributed under the terms of the Creative Commons Attribution License, which permits unrestricted use, distribution, reproduction and adaptation in any medium and for any purpose provided that it is properly attributed. For attribution, the original author(s), title, publication source (PeerJ) and either DOI or URL of the article must be cited.
License URL: https://creativecommons.org/licenses/by/4.0/

Keywords: Pneumonia, Effectiveness, Streptococcus pneumoniae, Epidemiology, Quasi-experimental study, Vaccination, Cause of death, Pneumococcal polysaccharide vaccine, Pneumococcal conjugate vaccine

Funding: Japan Agency for Medical Research and Development JP18fk0108050 and JP18fk0108066 Japan Science and Technology Agency (JST) CREST program JPMJCR1413 Telecommunication Advancement Foundation, and the Japan Society for the Promotion of Science (JSPS) KAKENHI 16KT0130, 17H04701, and 18H04895 MEXT Postgraduate Scholarship Programme Hiroshi Nishiura received funding support from the Japan Agency for Medical Research and Development (JP18fk0108050 and JP18fk0108066), the Japan Science and Technology Agency (JST) CREST program (JPMJCR1413), the Telecommunication Advancement Foundation, and the Japan Society for the Promotion of Science (JSPS) KAKENHI (16KT0130, 17H04701, and 18H04895). Sung-mok Jung received support from the MEXT Postgraduate Scholarship Programme. The funders had no role in study design, data collection and analysis, decision to publish, or preparation of the manuscript.

==============================
Background

It is plausible that the routine immunization among infants using pneumococcal conjugate vaccine 13 (PCV13) from 2013 and among the elderly using pneumococcal polysaccharide vaccine 23 (PPV23) from 2014 contributed to reducing the pneumonia mortality among the elderly in Japan. The present study aimed to estimate the causal effect of this vaccination on pneumonia mortality, using the available cause-of-death data and employing a difference-in-difference (DID) design.

Methods

Two types of mortality data, that is, prefecture-dependent and age- and gender-specific mortality data, from 2003 to 2017 were retrieved. We used mortality due to malignant neoplasm and heart disease as control groups and employed a DID design with an assumed parallel mortality trend between pneumonia and control group mortality since 2013 to estimate the causal effect of pneumococcal vaccination from 2014.

Results

Our estimation based on malignant neoplasm and heart disease as controls indicated that the reduced pneumonia mortality in 2017 owing to pneumococcal vaccination was as large as 41.9 (33.2, 50.6) and 31.2 (23.8, 38.6) per 100,000 individuals, respectively. The largest mortality reduction was observed for the oldest group (aged ≥90 years), especially among men.

Discussion

The pneumococcal vaccination program, perhaps mainly represented by high vaccination coverage of PCV13 among children and partly by PPV23 administration with low coverage among the elderly in Japan, was shown to have reduced the pneumonia mortality in the elderly at the population level.

Introduction

Japan, as a superaged nation, has a unique structure of causes of death (Arai et al., 2015). After successfully controlling infectious diseases following World War II, the country experienced a so-called “epidemiologic transition” with a dramatic change in the disease and cause-of-death structures (Omran, 2005). In the recent decade, the four major causes of death have been malignant neoplasm, heart disease, cerebrovascular disease (CVD), and pneumonia, all of which are most frequently seen in the elderly (Ministry of Health, Labour and Welfare, Japan, 2018a) (Fig. 1A). While the CVD mortality has followed a declining trend, perhaps reflecting gradual improvements in the diagnosis, treatment, and management of patients, the mortality from the other three major causes has steadily increased over time (Fig. 1B). These increases are considered to reflect the aging process as a whole in addition to aging within the elderly group. Unfortunately, from a clinical point of view, there has been no breakthrough in the prevention or treatment for these three diseases (Ebihara, 2017).

Figure 1 Mortality trends of major causes of death in Japan, 1947–2017.

(A) Long-term mortality trends for the four leading causes of death in Japan, that is, malignant neoplasm, heart disease, cerebrovascular disease, and pneumonia, are shown along with the long-term trend for tuberculosis mortality, which has substantially decreased since 1947. (B) Enlarged view of recent mortality trends for three major causes of death, that is, malignant neoplasm, heart disease, and pneumonia that mostly maintain a parallel monotonic increasing trend. In 2009, the subsidized immunization with pneumococcal conjugate vaccine (PCV) was initiated in a limited number of municipalities. In 2014, routine immunization with PCV began, targeting those who newly reached 65, 70, 75, 80, 85, 90, 95, or 100 years of age in the corresponding year from 2014 to 2018. The routine pneumococcal immunization of children started in 2013.

Nevertheless, it is noteworthy that an important change has occurred in reducing the risk of community-acquired pneumonia, especially pneumococcal pneumonia caused by Streptococcus pneumoniae, which accounts for 18.8% of the total cases (The Japanese Respiratory Society, 2017; Miyashita & Yamauchi, 2018). Following the subsidy-based exploratory immunization that began in 2009 at a municipality level with small coverages, in 2014 Japan began a routine immunization program with the pneumococcal polysaccharide vaccine 23 (PPV23) among the elderly (Naito, Yokokawa & Watanabe, 2018). In that program, elderly individuals aged 65 years or older were eligible for vaccination; owing to the policy intent to cover the entire elderly population within 5 years, those who newly reached the age of 65, 70, 75, 80, 85, 90, 95, or 100 years from 2014 to 2018 were invited to receive the immunization (Naito, Yokokawa & Watanabe, 2018). At yearly basis, crude vaccination coverage of PPV23 among elderly from 2014 to 2016 was estimated at 38.3%, 33.5%, and 37.8%, respectively (Ministry of Health, Labour and Welfare, Japan, 2018b). As of 2018, a vaccination coverage slightly above 40% has been achieved among the elderly (Naito, Yokokawa & Watanabe, 2018; Ministry of Health, Labour and Welfare, Japan, 2018b). In addition to vaccinating the elderly with PPV23, which is considered to protect vaccinated individuals from 23 types of pneumococci bacteria, the immunization of infants with the pneumococcal conjugate vaccine (PCV)7 was introduced as voluntary vaccination from late 2010 with the initially estimated coverage below 40%. Subsequently, public subsidy of PCV7 vaccination started in 2011 or 2012 depending on policy decision by local governments, and where subsidized, the vaccination coverage was elevated to be greater than 80% in the corresponding cities. In 2013, pneumococcal conjugate vaccine 13 (PCV13) replaced PCV7, and the vaccination with PCV13 became a part of routine immunization program. Regardless of PCV7 or PCV13, the first, second and third doses were given at 2 months old, 4 months old, and 6 month old, respectively, with an optional supplementary dose at 12–15 months old. Yearly vaccination coverage among children for three doses of PCV13, as calculated by the ratio of the total number of third doses to the demographic size of the subject age group, from 2013 to 2016 all exceeded 99.9% (i.e., 103.4%, 102.5%, 106.5%, and 99.9%, respectively), because people receiving this vaccination may have slightly extended to other (perhaps older) age groups (Ministry of Health, Labour and Welfare, Japan, 2018b). Considering the high vaccination coverage, this change has been expected to not only reduce the invasive pneumococcal disease (IPD) among infants but also reduce the transmission of these diseases from children to the elderly (Bogaert, De Groot & Hermans, 2004).

The routine administration of both these pneumococcal vaccines seemed likely to reduce the pneumonia mortality among the elderly in Japan. A published study in Canada indicated that the risk of death, frequency of complications, and length of hospital stay among pneumonia patients were all reduced by prior pneumococcal vaccination (Fisman et al., 2006). Moreover, PPV23 has been shown to prevent pneumonia in elderly people (Ortqvist et al., 1998). Nevertheless, a population-based cohort study involving hospital-admitted pneumonia patients in Canada demonstrated that, if admitted patients were vaccinated and followed up, PPV23 immunization was not associated with a reduced risk of death or further hospitalization (Johnstone et al., 2010). Another cohort study in Italy has shown that while PPV23 administration certainly improved the survival rate of elderly individuals, those who received PCV13 enjoyed a better survival rate (Baldo et al., 2016). Scientific evidence supporting the effectiveness of pneumococcal vaccines for preventing pneumonia death is still scarce, and it has not been directly assessed among the elderly in Japan.

A close look at the pneumonia mortality in Japan in recent years (Fig. 1B) reveals an overall stagnation with a potential signature of decline. Notably, aging progressed during this period both for the entire population and within the elderly population, and, in line with this, the mortality due to two other major causes, that is, malignant neoplasm and heart disease, steadily increased from 2003 to 2017. This observation may offer an important avenue by which to study the effectiveness of pneumococcal vaccination in Japan. The present study aimed to estimate the impact of pneumococcal vaccination on pneumonia mortality among the elderly, using the cause-of-death data in Japan and employing a quasi-experimental study design.

Materials and Methods

Mortality data

The present study was conducted using publicly available datasets on the cause-of-death in Japan from 2003 to 2017 (Ministry of Health, Labour and Welfare, Japan, 2018a). The time frame of 2003–2017 was specifically selected because the elderly aged 80 years or older were grouped together in the data from earlier years and also because the age-specific mortality trend appeared to be approximately linear during this period based on a visual assessment. Otherwise, the data were available in 5-year age bands. We restricted ourselves to analyzing the mortality trends of only three major causes of death, namely, malignant neoplasm, heart disease, and pneumonia, because these were expected to increase with the aging population and there have not been any breakthroughs other than pneumococcal vaccination that might abruptly reduce these mortalities.

We collected two types of mortality data. One was the prefecture-dependent mortality rate per 100,000 individuals from 2003 to 2017. For this, the cause-specific mortality rates for 47 prefectures were amassed. The other dataset was the age- and gender-dependent mortality rate per 100,000 individuals from 2003 to 2017. Ages were grouped in 5-year increments. Among the elderly, pneumonia was ranked within the top five causes of death from around the ages of 65–69 years old. Thus, our subject ages were split into six age-groups, that is, those aged 65–69, 70–74, 75–79, 80–84, 85–89, and ≥90 years, stratified by gender. Moreover, to validate our underlying assumption of parallel trend between pneumonia mortality and control disease selected either from malignant neoplasm or heart disease, we also collected the mortality data due to chronic obstructive pulmonary disease (COPD) and used it as the control during sensitivity analysis (see statistical analysis). To do so, the COPD-induced mortality data for the entire Japan from 2003 to 2017 was collected, and compared against pneumonia mortality.

Epidemiological analysis

The present study employed a difference-in-difference (DID) design, which is a quasi-experimental epidemiological study design that can be applied to observational data (Wing, Simon & Bello-Gomez, 2018). To purify the causal impact of pneumococcal vaccination on pneumonia mortality among the elderly, we compared the mortality trend of pneumonia death against the mortality of two control cause-of-death groups, that is, malignant neoplasm and heart disease. From a visual assessment of the data shown in Fig. 1B, the mortality of these three major diseases have increased roughly in parallel over the last decades. Exploiting this observation, we independently applied the DID design to both the prefecture and age/gender data.

We used g = 1,…, G to index cross-sectional units, such as geographic areas, a = 1,…, A to index age groups, and t = 2003,…, 2017 to indicate year. The term Ygtk represents the outcome (i.e., mortality) against the mortality of control group k for unit g in year t. For the analysis of prefectural data, we used the expected mortality value calculated as (1) E(Ygtk)=α0+α1(t−t0)+β(t−tv)(TgkPt)+γTgk+δPt,

to predict the pneumonia mortality and also the mortality of control disease k, where α0 and α1 describe the mortality of a control group that is assumed to share the same mortality trend in parallel with the pneumonia group over a set time period, t0 is the first year of our analysis (i.e., t0 = 2003), β is the causal effect of vaccination on pneumonia mortality, tv is the first year when vaccination starts to influence the dynamics (i.e., tv = 2013), γ is the group effect on pneumonia mortality, and δ is the time effect on both pneumonia and control group k that was introduced at the same time as the pneumococcal vaccination. Because routine pneumococcal immunization with PCV for infants began in 2013, tv was assumed to be 2013. Because Japan implemented vaccination spending 5 years to cover all corresponding birth cohorts of elderly, and due to gradual roll-out effect of indirect benefit, the causal effect in the present study is modeled as the rate of change per year (i.e., expressed as yearly reduction rate in pneumonia mortality). Tgk and Pt are dichotomous dummy variables, defined as (2) Tgk={1 for pneumonia group0 for control group,

and(3) Pt={1 for t≥20140 otherwise,

respectively.

Similarly, but separately, we analyzed the dataset of age- and gender-dependent mortality data. The term Ysatk represents the outcome (i.e., the nationwide mortality) of age a and gender s in year t against the mortality of control group k. We used the expected mortality value calculated as(4) E(Ysatk)=α0,sa+α1,sa(t−t0)+βsa(t−tv)(TgkPt)+γsaTgk+δsaPt,

to predict the pneumonia mortality and also the mortality of a control disease, where t0, tv, and dummy variables Tgk and Pt are the same as in Eq. (1). The difference between model (4) and model (1) is that the parameters, α0, α1, β, γ, and δ are all dealt with as a function of age a and gender s for control disease group k in model (4).

Statistical analysis

For each control disease, that is, malignant neoplasm and heart disease, we separately estimated the causal effect of pneumococcal vaccination β on pneumonia mortality for both models (1) and (4). As part of sensitivity analysis, an additional analysis was conducted using the COPD as a control disease, and we confirmed that the causal impact of pneumococcal vaccination on pneumonia mortality among elderly would be maintained. The tenth revision of the International Classification of Diseases (ICD-10) began to be strictly enforced in January 2017 with respect to the rule to specify the underlying cause of death (rather than identifying the resulting pathophysiological process as the cause of death) (Ministry of Health, Labour and Welfare, Japan, 2018a), and thus, the time-dependent decline in pneumonia mortality may potentially be explained by this change. To avoid misattributing that overlapping effect, we estimated the causal effect of vaccination by analyzing the mortality both with and without the 2017 data. The non-linear least sum of squares method was employed to fit models (1) and (4) to the corresponding data from 2003 to 2017. The 95% confidence intervals (CI) of parameters were computed by the profile likelihood method. All statistical data were analyzed using JMP version 12.0.1 statistical software (SAS Institute Inc., Cary, NC, USA).

Ethical considerations

The present study analyzed data is publicly available (Ministry of Health, Labour and Welfare, Japan, 2018a). As such, the datasets used in our study were de-identified and fully anonymized in advance, and the analysis of publicly available data without identity information does not require ethical approval.

Results

Table 1 shows the summary statistics of mortality for three causes of death, that is, pneumonia, malignant neoplasm and heart disease. All of them showed increasing trend with age. The gap of mortality by age was greatest for pneumonia, for example, while the pneumonia mortality among those aged 65–69 years were 43 and 11 per 100,000 for male and female, respectively, these figures among people aged 90 years and older were 2,883 and 1,438 per 100,000, respectively. We have compared correlations of mortality between pneumonia and control disease (i.e., malignant neoplasm or heart disease) by age group and gender (see Table S1). Comparing those correlations from 2003 to 2013 and from 2014 to 2017, we found that the parallel trend between pneumonia and control disease was maintained better prior to 2013, and subsequently the correlations were mostly diminished. During the latter part of the time period, the pneumonia mortality data were probably affected by vaccination.

Table 1 Summary statistics of mortality by cause of death, gender and age.

Disease	Gender	Age (years)	With 2017	Without 2017	
Min	Average	Max	Min	Average	Max	
Malignant neoplasm	Male	65–69	618	680	768	622	685	768	
Malignant neoplasm	Male	70–74	926	1,036	1,217	946	1,044	1,217	
Malignant neoplasm	Male	75–79	1,292	1,519	1,695	1,307	1,535	1,695	
Malignant neoplasm	Male	80–84	1,925	2,141	2,242	1,953	2,157	2,242	
Malignant neoplasm	Male	85–89	2,723	2,834	2,933	2,784	2,842	2,933	
Malignant neoplasm	Male	90 & over	3,309	3,431	3,603	3,309	3,430	3,603	
Malignant neoplasm	Female	65–69	286	297	315	286	297	315	
Malignant neoplasm	Female	70–74	388	421	458	395	424	458	
Malignant neoplasm	Female	75–79	546	613	659	567	618	659	
Malignant neoplasm	Female	80–84	842	906	965	863	910	965	
Malignant neoplasm	Female	85–89	1,217	1,297	1,387	1,237	1,303	1,387	
Malignant neoplasm	Female	90 & over	1,648	1,691	1,752	1,648	1,692	1,752	
Heart disease	Male	65–69	168	190	217	168	191	217	
Heart disease	Male	70–74	253	297	357	259	300	357	
Heart disease	Male	75–79	425	518	612	425	524	612	
Heart disease	Male	80–84	814	964	1,083	822	975	1,083	
Heart disease	Male	85–89	1,623	1,823	2,082	1,623	1,837	2,082	
Heart disease	Male	90 & over	3,407	3,664	3,935	3,407	3,677	3,935	
Heart disease	Female	65–69	53	69	83	53	677	83	
Heart disease	Female	70–74	100	1,374	172	108	136	172	
Heart disease	Female	75–79	2,185	287	356	225	292	356	
Heart disease	Female	80–84	508	640	759	514	649	759	
Heart disease	Female	85–89	1,167	1,396	1,624	1,167	1,411	1,624	
Heart disease	Female	90 & over	3,071	3,317	3,533	3,071	3,333	3,533	
Pneumonia	Male	65–69	43	63	72	57	65	72	
Pneumonia	Male	70–74	93	147	175	130	151	175	
Pneumonia	Male	75–79	211	357	431	285	367	431	
Pneumonia	Male	80–84	521	850	991	692	873	991	
Pneumonia	Male	85–89	1,231	1,870	2,199	1,577	1,916	2,199	
Pneumonia	Male	90 & over	2,883	4,147	4,764	3,421	4,237	4,764	
Pneumonia	Female	65–69	11	19	23	16	19	23	
Pneumonia	Female	70–74	24	46	58	39	47	58	
Pneumonia	Female	75–79	67	120	149	94	123	149	
Pneumonia	Female	80–84	187	320	389	252	330	389	
Pneumonia	Female	85–89	490	805	978	630	828	978	
Pneumonia	Female	90 & over	1,438	2,176	2,531	1,744	2,229	2,531	
Note:

Mortality per 100,000 individuals is shown. Minimum and maximum refer to the largest and smallest yearly mortality values from the year 2003 up to 2016 or 2017.

The estimated risk reductions of pneumonia mortality attributable to pneumococcal vaccination, using the mortality of malignant neoplasm or heart disease as the control, are shown in Fig. 2. In all analyses, the upper 95% CI values were less than 0, and a time-dependent increase in the causal reduction effect of pneumococcal vaccination on pneumonia mortality was observed. Table 2 shows estimates of the causal parameter β which indicates the yearly reduction rate of pneumonia mortality that can be attributed to pneumococcal vaccination (see Table S2 for other parameters). When the 2017 data were included, the analysis with malignant neoplasm or heart disease as a control indicated that the reduced pneumonia mortality in 2017 was as large as 41.9 (33.2, 50.6) or 31.2 (23.8, 38.6) per 100,000 individuals, respectively. When the 2017 data were excluded from the analyses, these mortality reductions did not deviate significantly and were estimated at 34.6 (22.0, 47.1) or 19.0 (8.3, 29.6) per 100,000 when malignant neoplasm or heart disease, respectively, was used as the control.

Figure 2 Causal effect of the pneumococcal vaccine as estimated from prefectural mortality data.

(A–D) The causal effect of the pneumococcal vaccine based on an analysis including (A–B) or excluding (C–D) the 2017 data and using malignant neoplasm (A and C) or heart disease (B and D) as the control. The vertical axis measures the causal effect that is calculated by β (Year–2013) described from Eq. (1), where β is the yearly reduction in the pneumonia mortality per 100,000 individuals. The solid line shows the estimated causal effect, while the dashed and dash-dotted lines show the 95% upper and lower CI, respectively.

Table 2 Estimates of the age-dependent causal effect parameter of pneumococcal vaccination by age group and gender in Japan.

Age (years)	With 2017 data	Without 2017 data	
Malignant neoplasm	Heart disease	Malignant neoplasm	Heart disease	
Male	Female	Male	Female	Male	Female	Male	Female	
65–69	16 (−28, 59)	1 (−19, 21)	2 (−37, 41)	3 (−22, 27)	22 (−42, 86)	3 (−27, 32)	6 (−50, 63)	5 (−30, 40)	
70–74	23 (−20, 67)	5 (−15, 24)	2 (−37, 41)	5 (−19, 30)	34 (−30, 98)	8 (−22, 37)	9 (−47, 66)	9 (−26, 44)	
75–79	33 (−11, 75)	3 (−16, 23)	−9 (−48, 30)	8 (−16, 33)	53 (−11, 117)	8 (−21, 37)	4 (−53, 60)	16 (−19, 51)	
80–84	−31 (−74, 12)	−18 (−38, 2)	−42 (−81, −3)	7 (−18, 31)	−8 (−72, 56)	−11 (−41, 18)	−20 (−76, 36)	21 (−14, 56)	
85–89	−143 (−186, −99)	−60 (−80, −40)	−98 (−137, −59)	−12 (−37, 12)	−116 (−180, −52)	−49 (−79, −20)	−52 (−108, 4)	12 (−23, 47)	
90 & over	−357 (−401, −314)	−199 (−218, −179)	−276 (−315, −237)	−137 (−162, −113)	−339 (−403, −275)	−186 (−215, −156)	−226 (−282, −170)	−105 (−140, −70)	
Note:

Figures should be interpreted as the yearly rate of reduction in pneumonia mortality per 100,000 individuals. Malignant neoplasm and heart disease were used as control groups. Upper and lower 95% confidence intervals, derived from profile likelihood, are shown in parenthesis.

Comparisons of the expected and observed mortality values are shown in Fig. 3, and parameter estimates of the causal model are shown as Table 3. The observed values are expressed as the distribution of mortality in the 47 prefectures (see Table S3 for the observed values of the entire Japan). Both control diseases had a pattern of monotonic increase in the mortality over time for both the expected and observed mortality; they shared this increasing trend with pneumonia mortality through 2013. However, an abrupt decline in the pneumonia mortality was observed beginning in 2013. This phenomenon was also captured by the simple DID model described above; together, they support the causal effect of pneumococcal vaccination on the decline in pneumonia mortality.

Figure 3 Comparison of observed and predicted mortality from the analysis of prefectural data.

(A–B) Mortality calculated using malignant neoplasm (A) or heart disease (B) as the control group. Gray circles represent the predicted mortality of malignant neoplasm (A) or heart disease (B), squares show the median of malignant neoplasm (A) or heart disease (B) mortality among the 47 prefectures, and error bars indicate the minimum and maximum values. The vertical axis represents the mortality per 100,000 individuals. Black circles represent the predicted pneumonia mortality, triangles show the median of pneumonia mortality among the 47 prefectures, and error bars indicate the minimum and maximum values. The arrow indicates the year 2013 in which routine immunization started among elderly.

Table 3 Estimates of the regression parameters for the causal effect model of pneumococcal vaccination applied to prefectural data in Japan.

	With 2017 data	Without 2017 data	
Malignant neoplasm	Heart disease	Malignant neoplasm	Heart disease	
α0	267 (264, 271)	141 (138, 144)	268 (264, 271)	142 (138, 145)	
α1	4 (3, 4)	3 (3, 4)	4 (3, 4)	3 (3, 4)	
β	−10 (−13, −8)	−8 (−10, −6)	−9 (−12, −6)	−5 (−7, −2)	
γ	−187 (−190, −183)	−58 (−61, −55)	−187 (−191, −184)	−59 (−62, −56)	
δ	1 (−5, 6)	−5 (−9, 0)	−0 (−6, 6)	−7 (−12, −1)	
Note:

Figures should be interpreted as the yearly rate of reduction in pneumonia mortality per 100,000 individuals. Malignant neoplasm and heart disease were used as control groups. Upper and lower 95% confidence intervals, derived from profile likelihood, are shown in parenthesis.

The estimated causal effects of pneumococcal vaccination in reducing age- and gender-specific pneumonia mortality, using malignant neoplasm and heart disease as the control group based on analyses with and without the 2017 data, are shown in Fig. 4. We did not identify statistically significant reductions in the pneumonia mortality among those aged from 65 to 79 years for either males or females in any of our analyses. In the older age groups, vaccine-induced reductions in the pneumonia mortality were seen, and the estimated effect size grew with the subject age, especially among men. In the analyses that included the 2017 data and used malignant neoplasm or heart disease as the control, the pneumonia mortality in elderly males aged ≥90 years was reduced every year from 2014 by 357.4 (95% CI [314.0–400.7]) or 276.3 (95% CI [237.3–315.2]) per 100,000 individuals, respectively, as a function of time since vaccination effect. Similarly, the yearly reduction in pneumonia mortality among females was estimated at 185.5 (95% CI [156.2–214.9]) or 105.2 (95% CI [70.2–140.3]) per 100,000 individuals when malignant neoplasm or heart disease, respectively, were used as the control.

Figure 4 Age- and gender-specific causal effect of pneumococcal vaccination.

(A–D) The age- and gender-specific causal effect of the pneumococcal vaccine based on an analysis including (A–B) or excluding (C–D) the 2017 data and using malignant neoplasm (A and C) or heart disease (B and D) as the control. The causal effect in decreasing pneumonia mortality is marked as a dashed line, and the 95% confidence intervals of the causal effect are presented using error bars.

There was a good overall level of agreement between the observed and expected mortality data with respect to age and gender for pneumonia (Figs. 5A, 5B, 5E, and 5F) and the control causes of death (Figs. 5C, 5D, 5G, and 5H). In different age and gender groups, a different linear trend of mortality was shared between pneumonia and each control disease beginning in 2013, after which a clear decline in pneumonia mortality was identifiable, especially among those aged ≥90 years.

Figure 5 Comparison of observed and predicted age- and gender-specific mortality.

(A–B) Age-stratified pneumonia mortality in males (A) or females (B), using malignant neoplasm as the control for calculating predicted mortality. Each line represents the modelled (predicted) mortality. Each sign represents the observed mortality data. Since 2014, the year routine immunization was introduced, the observed pneumonia mortality shows a decreasing trend. (C–D) Malignant neoplasm mortality by age in males (C) and females (D). (E–F) Age-stratified pneumonia mortality in males (E) or females (F), using heart disease as the control for calculating predicted mortality. (G–H) Heart disease mortality by age in males (G) or females (H).

As part of sensitivity analysis, Table 4 examines the causal effect when 2014 was used, rather than 2013, as the year of change point, considering that the completion of vaccination among elderly is 2014, while that of children was 2013. The yearly reduction rate in pneumonia mortality was estimated to be greater for this model, compared with 2013 model. Causal parameter estimates in the 2014 model were greater than those of 2013 model, because substantial number of elderly started to be vaccinated within 2013 and corresponding herd immunity effect was perhaps captured by the 2014 model.

Table 4 Estimates of the causal effect of pneumococcal vaccination on pneumonia death by age group and gender.

Age (years)	With 2017 data	Without 2017 data	
Malignant neoplasm	Heart disease	Malignant neoplasm	Heart disease	
Male	Female	Male	Female	Male	Female	Male	Female	
65–69	20 (−52, 92)	1 (−32, 34)	2 (−64, 68)	3 (−38, 44)	34 (−90, 158)	4 (−53, 62)	9 (−104, 123)	7 (−63, 77)	
70–74	24 (−48, 96)	4 (−29, 37)	0 (−66, 67)	6 (−35, 47)	41 (−83, 165)	8 (−49, 66)	13 (−101, 126)	12 (−58, 82)	
75–79	42 (−30, 114)	5 (−29, 38)	−13 (−79, 53)	11 (−30, 52)	84 (−40, 208)	13 (−45, 70)	8 (−106, 121)	24 (−46, 94)	
80–84	−38 (−110, 34)	−25 (−58, 8)	−57 (−124, 9)	8 (−33, 49)	0 (−124, 124)	−15 (−73, 42)	−23 (−137, 90)	33 (−37, 103)	
85–89	−195 (−267, −123)	−84 (−118, −51)	−140 (−206, −73)	−18 (−59, 23)	−167 (−291 −43)	−74 (−133, −17)	−73 (−186, 41)	24 (−46, 94)	
90 & over	−491 (−563, −419)	−275 (−308, −242)	−376 (−442, −310)	−183 (−224, −142)	−510 (−634, −386)	−283 (−340, −226)	−322 (−436, −209)	−146 (−216, −76)	
Prefecture	2 (−4, 8)	−3 (−8, 2)	1 (−5, 8)	−6 (−11, 0)	
Note:

Figures should be interpreted as the yearly rate of reduction in pneumonia mortality per 100,000 individuals. Malignant neoplasm and heart disease were used as control groups. Except for the last row, age-dependent model results are shown, while the last row give estimates from prefecture-dependent model. Upper and lower 95% confidence intervals, derived from profile likelihood, are shown in parenthesis.

The mortality of COPD was used as the control as part of sensitivity analysis. As was also the case for malignant neoplasm and heart disease, the overall parallel trend between COPD and pneumonia mortality prior to vaccination, and an assumed causal impact of vaccination on mortality were reflected in the prediction using model (1) and qualitatively captured the observed patterns for the entire Japan (see Fig. S1). The maximum likelihood estimate of causal parameter remained to be negative (Table S4), that is, (−2 (95% CI [−5–1]), not much deviated from the results using other control diseases (Table 3).

Discussion

The present study estimated the causal effect of pneumococcal vaccination on pneumonia mortality among the elderly in Japan. A DID study design, which is a quasi-experimental epidemiological research design, was employed, using the time-dependent mortality of malignant neoplasm and heart disease as the control groups because these causes of death demonstrated a monotonic increase induced by population aging and shared this trend with pneumonia mortality at least through 2013. The subsequent abrupt decline in pneumonia mortality was captured through the analysis of prefectural data as well as of age- and gender-specific nationwide data. Overall, the pneumonia mortality in 2017 was considered to have been reduced by 20–40 per 100,000 individuals owing to pneumococcal vaccination, with the largest causal effect occurring among the oldest group, aged ≥90 years. To our knowledge, the present study is the first to have assessed the combined causal impact of two pneumococcal vaccination programs, that is, PCV13 among children and PPV23 among the elderly, in reducing pneumonia mortality in the elderly as evaluated at the population level. The vaccination coverage of PCV13 among children has achieved over 95%, while that among elderly with PPV23 has remained less than 50%, implying that the large population impact may be represented by herd immunity due to PCV13.

It is remarkable that an explicit mortality reduction was identified by this quasi-experimental study design. Our analyses revealed that larger causal impacts of pneumococcal vaccination on pneumonia mortality were observed in the more recent years. Although this causal impact was not clearly identified among elderly individuals aged from 65 to 79 years, a clear impact was seen for people in their 80s and 90s. The question of whether PPV23 administration results in a visible causal impact among elderly people has been controversial: while protection against IPD was identified (Cadeddu et al., 2012) and a certain protection against pneumococcal pneumonia was reported (Chidiac & Ader, 2009; Falkenhorst et al., 2017), the effectiveness of PPV23 was smaller than that of PCV13 (Baldo et al., 2016) and sometimes was only minimal (Johnstone et al., 2010). However, the indirect effect of PCV13 vaccination in infants and children on outcomes in adults and the elderly has been reported (Simonsen et al., 2014; Regev-Yochay et al., 2017; Tsaban & Ben-Shimol, 2017); this practice of pediatric vaccination clearly provides a herd immunity effect among the elderly (Haber et al., 2007; Simonsen et al., 2011). Although our study endorses those findings in other countries including North America, because of the population-based design of this work, we were not able to disentangle the causal effects attributed to the two vaccines. Considering limited coverage of PPV23 and high coverage of PCV13, it is likely that the observed reduction in pneumonia mortality among elderly can be attributed to herd immunity impact by PCV13. In fact, looking at the most vulnerable group aged 90 years and older (Fig. 5A), the decline in pneumonia mortality has started to take place since subsidy among infants from 2011, and the gradual decline afterward is consistent with the gradual increase in vaccinated children with PCV7 or PCV13. On the other hand, we do not identify visible impact of PPV23 introduction from 2013. However, due to similar timing of introduction for both vaccines, the observed effect can be still be partially attributed to not only PCV13 but also PPV23, and the present study was not able to purify the unique effect of PCV13 alone from the mixed population impact of PCV13 and PPV23.

On the basis of the nature of this study’s DID design, other plausible explanations for the time-dependent decline in pneumonia mortality were mostly excluded. Although the DID design required the assumption of proportional trends among the pneumonia, malignant neoplasm, and heart disease mortalities, the reasonable fits produced by these data justified the use of these control diseases. Furthermore, the monotonic time-dependent increase was mechanistically explainable based on progressive population aging. Because the 2017 data could have been impacted by ICD-10’s rule to identify the underlying cause of death, we conducted analyses both with and without the 2017 data, but the overall qualitative patterns of our findings were not altered by the exclusion. This finding supports our conclusion that the abrupt decline in pneumonia mortality in 2017 is not fully explained by the ICD-10 change.

In Japan, the number of individuals who “died of old age” (i.e., natural death from an ailment associated with aging) has steadily increased. It is possible that the category “died of old age” absorbed the apparent reduction in pneumonia deaths in recent years. However, that possibility leaves the question of why only pneumonia deaths, not those due to neoplasm or heart disease, were selectively miscategorized. Our use of malignant neoplasm and heart disease as controls in the DID design helped to eliminate several concerns that the recent reductions in pneumonia mortality could be attributed to causes other than the two pneumococcal vaccination programs.

The present study using mortality as the outcome succeeded in demonstrating that even this type of quasi-experimental study can offer a vivid effect size estimate of pneumococcal vaccination programs at the population level. To truly estimate the individual-based causal effect of vaccination, it is critical to conduct a prospective study. Such a study should ideally control the exposure (i.e., frequency of contact) to children, so that the indirect effect of PCV13 vaccination may be separated from the direct benefit of PPV23 vaccination.

Five limitations of this study should be noted. First, control groups other than malignant neoplasm and heart disease deaths were not explored. It is possible that a more monotonically increasing control (e.g., mortality following bone fracture) could be conceived and compared against pneumonia mortality; however, we had access to only major causes of death by prefecture and by age and gender. Second, the cause of death category of pneumonia includes not only community-acquired pneumonia but also hospital-acquired infections and aspiration pneumonia. The observed reduction in pneumonia mortality was not verified to have been caused by a specific reduction in mortality from pneumococcal pneumonia, and moreover, possible improvement in prognosis could have been absorbed as our causal effect of vaccination. Nevertheless, even provided that we explored more detailed etiology of pneumonia, the time trend might not be easily explainable. Third, the effectiveness estimate may be quantitatively associated with the vaccination coverage both in children and the elderly; thus, variations in the causal impact by prefecture could possibly be explained by differences in vaccination coverage. Unfortunately, we were not able to conduct an analysis of this possibility owing to limitations with our data. Fourth, PCV7 vaccination was introduced as voluntary vaccination in 2010, and the subsidy-based program started in 2011 at the municipal level; either or both could have affected the pneumonia mortality. However, coverage with the PCV7 vaccination remained very low in 2014. Similarly, the overall vaccination coverage from the subsidy-based program remained very low, not even reaching a few percent. Only in and after 2014 was the pneumococcal vaccination coverage of both children and the elderly elevated to substantial levels. Fifth, we did not have an access to the temporal datasets of IPD among children over time. It is fruitful to estimate the causal impact of pneumococcal vaccination among children, looking into the corresponding data of IPD.

To confirm our findings, a precise estimation of the causal effect of pneumococcal vaccination among the elderly must be determined by conducting a study with a prospective design. Nevertheless, by using only the available census-based observational data, we showed that a DID design can be exploited to examine the population impact of pneumococcal vaccination on pneumonia mortality among the elderly. We believe that our approach will shed light on the assessment of the combined effectiveness of pneumococcal vaccinations in Japan.

Conclusions

The present study employed a DID design using the time-dependent mortality of malignant neoplasm and heart disease as control groups to estimate the causal effect of pneumococcal vaccination on pneumonia mortality among the elderly in Japan. From 2014, an abrupt decline in pneumonia mortality was seen; specifically, the mortality was reduced by 20–40 per 100,000 individuals because of pneumococcal vaccination, with the largest causal effect among the oldest group aged 90 years and older.

Supplemental Information

Supplemental Information 1 Supplementary Figure S1.

Comparison of observed and predicted mortality of pneumonia and chronic obstructive pulmonary disease (COPD) from 2003-2017, Japan. Predicted mortality for the entire Japan was calculated, using the COPD mortality as the control group. The vertical axis represents the mortality per 100,000 individuals. Each line represents the predicted mortality, while each mark represents the observed data.

Click here for additional data file.

Supplemental Information 2 Supplementary Table 1. Correlation between pneumonia and control diseases (i.e., malignant neoplasm and heart disease) by age and gender.

Click here for additional data file.

Supplemental Information 3 Supplementary Table 2. Summary of causal model parameters for the analysis by age-group and gender.

Click here for additional data file.

Supplemental Information 4 Supplementary Table 3. Summary of observed and predicted mortality from the analysis of prefectural data.

Click here for additional data file.

Supplemental Information 5 Supplementary Table 4.

Supplementary Table 4. Estimates of the regression parameters of the causal effect model, comparing pneumonia mortality with that of chronic obstructive pulmonary disease (COPD) in Japan.

Click here for additional data file.

Additional Information and Declarations

Competing Interests

Author Contributions

Data Availability

Hiroshi Nishiura is an Academic Editor for PeerJ.

Sung-mok Jung performed the experiments, analyzed the data, prepared figures and/or tables, authored or reviewed drafts of the paper, approved the final draft.

Hyojung Lee performed the experiments, analyzed the data, prepared figures and/or tables, authored or reviewed drafts of the paper, approved the final draft.

Hiroshi Nishiura conceived and designed the experiments, performed the experiments, analyzed the data, contributed reagents/materials/analysis tools, authored or reviewed drafts of the paper, approved the final draft.

The following information was supplied regarding data availability:

The data is openly accessible (publicly released by a third-party). References to that data are included in the article:

https://www.mhlw.go.jp/toukei/list/81-1a.html.

https://www.mhlw.go.jp/topics/bcg/other/5.html.

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
