# Peer review of "The impact of pneumococcal vaccination on pneumonia mortality among the elderly in Japan: a difference-in-difference study"

_PeerJ, doi:10.7717/peerj.6085_

## Round 0.1 · original submission · Major Revisions

As you can see, your submission triggered contrasting opinions with Reviewer #1 suggesting rejection while Reviewer #2 suggested minor revision. Having read your submission my-self with attention, and taking the comments of the reviewers in consideration, I believe that your text may, possibly, be amended to meet most of the criticisms raised. However, it will essential that you submit a detailed rebuttal in which you explain how you dealt with each of the reviewers' comments and indicate how and where the submission has been modified (or not). Please, be aware that your revised submission will undergo a new round of review by the same or by different reviewers. I therefore cannot make any commitment here about final acceptability of your work for publication in the Journal.

Reviewer 1 ·

Basic reporting

This clearly written paper is on a very important and unsettled question. The analysis and the figures are clearly laid out and described. The authors describe the existing literature on the effectiveness of PPCV13 and PPV23 and place their paper in this literature.

Experimental design

The research question is well defined and the analysis is well described.

Validity of the findings

All my concerns have to do with the analysis. In order of importance.

1. The authors should do more to assess the parallel trends assumption. They are finding very big effects that are not visible in Figure 1B. These may be the result of diverging trends. Any progress the authors make on directly assessing the parallel trends assumption such as presenting the differences in mortality rates between pneumonia and the control groups would strengthen the analysis.

2. Why are the regressions specified to capture a break in the trend rather than a sharp change in the level? Is it due to the slow roll out of the vaccine?

3. More detail on vaccine uptake would really strengthen the paper. This would allow the readers to scale the effects. If uptake is as low for the relevant population as suggested by the overall numbers in the paper, then treatment on the treated is much larger than the effects documented in the paper. Unless of course there are large herd effects.

4. It may be worth implementing a difference in difference analysis that leverages the age specific recommendations for the vaccine. This might provide a cleaner comparison or at least one more counterfactual. People 60-64 might provide a better counterfactual to those 65-69 than the cause of death counterfactual.

5. Regression specification (1) allows for a level change (P_t) at the time of the roll out of the second vaccine. This might absorb some of the treatment effect.

Reviewer 2 ·

Basic reporting

Overall, the article was clear throughout and structured in an easy to read format. The literature review seemed lacking as there are many articles on the effectiveness of the pneumococcal vaccine and not many are mentioned. There are a few additions that I think would improve the paper:

1. I would also like to see tables with basic summary statistics of the data. I'd like to know the underlying rates of pneumonia mortality by gender and age to be able to have a reference point for the estimates that you find. They seem like very large numbers (perhaps implausibly large... but it is hard to know without the summary statistics.)

2. I would like to see tables of the results. While I like the results figures, it would be nice to see the coefficients and standard errors in an easy to read table format that also includes whether standard errors are robust to heteroskedasticity or clustered.

3. I would include the data from the observed values of figure 3 along with a line to mark the introduction of the pneumococcal vaccine to motivate the DID design. I'm not sure that the predicted values add much to the graph or reporting of the results.

Experimental design

I agree that the rollout of vaccination allows for a nice difference in differences design. I question whether the choice of heart disease and malignant neoplasms are appropriate control groups. The authors defend this choice as these are prevalent causes of death. However, it seems that something like pneumonia might have more seasonal and year to year variation than heart disease or malignant neoplasms. I'd think that respiratory illness would be a more appropriate control group.

I am also not sure why the results only focus on the elderly given the vaccination of children beginning in 2013. It seems that childhood mortality would also be an interesting outcome measure. I am also not convinced that 2013 is the appropriate year to use for the post treatment variable with elderly outcomes since elderly vaccination did not occur until 2014. It would at least be nice to see that the results are robust to using 2014 as the post year.

I am assuming that the data are only available in 5 year age bands. If not, then age should be incorporated more fully in the DID design since different ages were vaccinated at different times.

Finally, it would be nice to show vaccination rates over time to provide support for the DID design. Given some vaccinations started prior to 2014 and the vaccinations were phased in gradually after 2014, it is not clear that there really is a clear jump in vaccinations during the post period.

Validity of the findings

No further comments. See above for some concerns about the methodology which would cause concerns for the validity of the findings.

---

## Round 0.2 · Major Revisions

As you can see, your revised version convinced one of the reviewers but not the other one. This prevents acceptance at this stage because of questions about the validity of the findings. I, therefore, would like you to seriously consider the remarks of the reviewer and to come with a detailed rebuttal. You may agree or disagree with the reviewer, but the rebuttal (and, possibly, a further revised version) must be convincing. I hope you will be able to do so.

Reviewer 1 ·

Basic reporting

Clearly written.

Experimental design

Well described and sensible

Validity of the findings

Look reasonable.

Reviewer 2 ·

Basic reporting

I appreciate the summary table of underlying mortality rates and seeing the results in table form. I think this makes the paper easier to read and evaluate. I also appreciate some data on vaccination rates, although more details across age and time would have been nicer.

Experimental design

I am still concerned that the control group is heart disease and malignant neoplasms. The authors did not try to use another control such as respiratory disease… or perhaps deaths in another country that does not have a pneumococcal vaccination program (see Vivek Charu et al, 2011 for a cross country design). Given the seasonal nature of pneumonia, a few years off trend from the more steady trend of malignant neoplasm is not surprising.

Validity of the findings

In light of reporting on the vaccination levels and lack of robustness with other control groups, I am concerned about the validity of the findings.

The vaccination levels suggest small uptake in the elderly which makes any direct effects seem unlikely (especially because the vaccine only protects against a fraction of pneumonia cases to begin with). Moreover, vaccination rates of the children have not been given prior to 2013 to support the story that vaccination increased in 2013. More details of the child policy need to be explained (which ages were vaccinated? And what do vaccination rates look like prior to the policy). If we are to believe the results, this is really a story of a herd effect that gives protection to the elderly from the vaccination of youth. The elderly vaccination rates changed very little suggesting that any effect can’t be driven by direct vaccination (unless the change in rates was significantly bigger for the 80+ age group than the numbers reported on the whole.) Thus, I think the paper should be written more clearly to express this idea.

In addition, given that the effects must be driven by the youth, it seems that we should expect to see a jump in the data at 2013, not a slow decline in mortality from 2014. It is not clear that we see that in the data. Again, it would help to understand the child policy better and the evolution of vaccination rates among children to understand what we expect to happen in the data.

Comments for the author

I believe that the results cannot reflect effects of elderly vaccination given small changes in vaccination rates and large effect sizes. Thus, the paper needs to be written more clearly to reflect an argument of herd effects. In addition, I think more evidence of vaccination rates among children across time and age need to be presented to make the argument for herd effects.

---

## Round 0.3 · accepted · Accept

Your revised version was considered more acceptable and worthy of publication.

Reviewer 2 ·

Basic reporting

The paper was clear and easy to read. No further comments.

Experimental design

No further comments.

Validity of the findings

I think the claims seem more reasonable in this version of the paper.